# Predicting the Future Geographic Distribution of the Traditional Chinese Medicinal Plant *Epimedium acuminatum* Franch. in China Using Ensemble Models Based on Biomod2

**DOI:** 10.3390/plants14071065

**Published:** 2025-03-30

**Authors:** Zhiling Wang, Zhihang Zhuo, Biyu Liu, Yaqin Peng, Danping Xu

**Affiliations:** College of Life Science, China West Normal University, Nanchong 637002, China; wangzhiling995@foxmail.com (Z.W.); zhuozhihang@cwnu.edu.cn (Z.Z.); biyuliuql@foxmail.com (B.L.); pengyaqin2023@foxmail.com (Y.P.)

**Keywords:** *Epimedium acuminatum* Franch., Chinese medicinal plants, potential distribution, Biomod2, suitable habitat

## Abstract

This study employs the Biomod2 model, along with 22 bioclimatic variables, to predict the geographic distribution of the medicinal plant *Epimedium acuminatum* Franch. for the current period and three future timeframes (2050s, 2070s, and 2090s). Ultimately, 11 key environmental variables were identified as critical for assessing the habitat suitability of the medicinal plant. These include the annual mean temperature (Bio 1), isothermally (Bio 3), temperature seasonality (Bio 4), maximum temperature of the warmest month (Bio 5), minimum temperature of the coldest month (Bio 6), mean temperature of the driest quarter (Bio 9), mean temperature of the coldest quarter (Bio 11), precipitation of the driest quarter (Bio 17), elevation (Elev), aspect, and slope. The results indicate that the current high suitability areas are primarily distributed across Yunnan, Chongqing, Sichuan, Hunan, Guangxi, and Hubei provinces. In the future, the extent of high suitability areas is expected to increase. This study aims to provide a theoretical reference for the conservation of *E. acuminatum* genetic resources from a geographic distribution perspective.

## 1. Introduction

Plants have historically served as the primary source of medicinal compounds worldwide and continue to play a crucial role in pharmaceutical research and development [1]. Notably, natural products contribute to approximately 50% of all clinically utilized drugs, with higher plants accounting for 25% of this total [2]. Despite advancements in modern medicine, a substantial proportion of clinically approved therapeutics still originate from bioactive compounds extracted from medicinal plants [3]. Medicinal plants are defined as those containing bioactive constituents with therapeutic or preventive potential against various diseases [4]. In many developing countries, medicinal plants remain integral to primary healthcare systems due to their affordability and accessibility. It is estimated that approximately 80% of the population in several developing nations, including South Africa, relies on traditional medicines for healthcare needs [2].

*Epimedium* is the largest herbaceous genus in the Berberidaceae (barberry) family. Plants in this genus have significant medicinal value and exhibit a high degree of species diversity [5,6,7]. Among them, *Epimedium acuminatum* Franch. is a representative species of the genus *Epimedium* in the southwest region of China [8]. It has several common names including barrenwort, bishop’s hat, and fairy wings. Over 2000 years ago, it was recorded as a tonic in the <Shennong Ben Cao Jing> and is also one of the earliest medicinal materials included in the modern <Chinese Pharmacopoeia> [9]. The 1988 edition of the Guizhou Province Traditional Chinese Medicine Standards also includes it as a medicinal material [10], with the primary active components being flavonoids [11]. *Epimedium* vulgare is a commonly used bulk Chinese medicine. The whole plant has medicinal value, and the roots and leaves are especially valuable [12]. Traditional Chinese medicine believes that it has the efficacy of strengthening muscles and bones, dispelling wind-dampness, and tonifying the kidney and strengthening the yang. In modern pharmacological research, it also has the effect of regulating immune function, increasing coronary flow, improving blood circulation of the heart and kidney, promoting nucleic acid metabolism, and anti-cancer and anti-aging properties [13,14]. Currently, research on *E. acuminatum* mainly focuses on the analysis and extraction of medicinal components, propagation, and photosynthetic physiological ecology [15,16], with very limited studies on its geographical distribution. It is worth noting that the cultivation techniques for *E. acuminatum* are still not fully developed, and the acquisition of medicinal materials largely relies on wild resources. This may lead to the depletion of wild resources before its medicinal value is fully realized [17]. Against this background, the potential impact of climate change on plant distribution has prompted research on predicting species’ geographic distribution changes, with species distribution models (SDMs) becoming a core tool in this field.

Climate change can have interconnected effects on many environmental conditions. It may lead to varying degrees of contraction, expansion, or alteration of the geographical range of medicinal plants [18]. This process may pose a potential threat to some species [19,20,21]. Species distribution models (SDMs) aim to estimate the presence of target species within a specific range [22]. They are important tools for developing strategies and policies for the management and sustainable, equitable use of biological resources [23]. Many studies use SDMs to predict species distribution changes caused by environmental changes, and there are also many different branches of these models [24]. Due to the diversity of concepts and algorithms, different models have their own limitations. Using a single model may lead to biased predictions because of a lack of understanding of the true relationship between species and environmental changes [25]. To improve the predictive accuracy of models, it is possible to combine multiple individual SDMs. This approach can leverage the strengths of different models while reducing their individual inaccuracies [26]. Biomod2 is built on this basis. In addition to the advantages mentioned above compared to single models, the use of Biomod2 further enhances the reliability and scientific accuracy of medicinal plant distribution predictions. This approach allows for a more comprehensive assessment of the impact of environmental variables on the species’ suitable habitat. Biomod2 is a modeling platform based on R software (R 4.3.2) [27,28]. Initial conditions and model parameters can be customized to obtain more accurate results [29]. In the Biomod2 integration methods for individual models, there are two types: EMca (Ensemble Model-CA) and EMwmean (Ensemble Model-Weighted Mean) [30,31,32,33]. EMca integrates models by calculating the category distributions of different models, while EMwmean combines model outputs using a weighted average approach. Both methods aim to enhance the accuracy and robustness of the predictions. EMca (Ensemble Model-CA) integrates the predictions of different models into a final classification result, typically determining the category of each pixel through a voting mechanism. EMwmean (Ensemble Model-Weighted Mean) calculates the weighted average of predictions from all models, with weights assigned based on the performance of each model [34].

This study aims to predict the potential geographic distribution patterns of *E. acuminatum* under current and future climatic conditions using the Biomod2 modeling platform by integrating multiple species distribution models with 19 bioclimatic variables and three topographic variables. By quantitatively assessing the impact of environmental changes on the suitable habitat of this species, this study not only elucidates its habitat dynamics and future distribution trends but also provides a theoretical basis for the scientific conservation and sustainable utilization of its wild populations. The findings are expected to contribute to the development of medicinal plant resource management and conservation strategies, thereby enhancing the significance of *E. acuminatum* in biodiversity conservation and the sustainable development of medicinal plant resources.

## 2. Results

### 2.1. Evaluation of SDMs and Environmental Variables

Using the filtered geographic distribution data for *E. acuminatum*, 10 models were developed. The TSS, Kappa, and AUC values for the two background points (PA1 and PA2) are presented in Table 1. Among them, RF showed the best overall performance, while GAM, GBM, ANN, FDA, MARS, GLM, and CTA all met the screening criteria outlined in Section 4.3 (TSS > 0.7, Kappa > 0.65, and AUC > 0.9). The remaining models, SRE (TSS = 0.699 for PA2) and MaxEnt (NA), did not pass the accuracy assessment and were excluded. In this context, “NA” indicates that the model data are incompatible and no values were generated. Subsequently, the eight remaining individual models were re-modeled using the two methods (EMca and EMwmean). Appendix A shows the TSS, Kappa, and AUC values for these ensemble models, both of which demonstrate excellent predictive performance.

This study employed Pearson correlation coefficients to assess the relationships among 19 bioclimatic variables (Figure 1). The PCA plot in Appendix A offers a clearer visualization of the contributions of these variables. Based on Pearson correlation coefficient criteria, climate factors more aligned with the biological characteristics of *E. acuminatum* were retained. Considering the contribution rates of each climate variable (Table 2), eight climate variables and three topographic factors were selected: annual mean temperature (Bio 1), isothermally (Bio 3), temperature seasonality (Bio 4), maximum temperature of the warmest month (Bio 5), minimum temperature of the coldest month (Bio 6), mean temperature of the driest quarter (Bio 9), mean temperature of the coldest quarter (Bio 11), precipitation of the driest quarter (Bio 17), along with elevation (Elev), aspect, and slope. Among these, Bio 6, Bio 11, and Bio 4 had the highest contribution rates at 64.80%, 36.80%, and 33.38%, respectively.

### 2.2. Response Curve Analysis (RF)

Based on the above results, it can be concluded that RF performs the best across all evaluation criteria. The TSS, Kappa, and AUC values demonstrate, from the perspectives of model prediction accuracy, reliability, and performance, that RF is more suitable for modeling this species. Thus, this study used RF to perform logistic regression analysis on the presence probability of *E. acuminatum* with the 11 selected bioclimatic variables and created univariate response curves (Figure 2). This figure visually demonstrates how presence probability varies with different variables, with the *x*-axis representing the variables and the *y*-axis showing the presence probability of *E. acuminatum*.

In Figure 2, the relationship is depicted using red lines for PA1 and blue lines for PA2. Within each color, the lines represent RUN1 (initial values) and FULL (mean values). A variable is deemed beneficial for the target species if the presence probability is ≥0.6. For the aspect variable (Figure 2a), the trends for PA1 and PA2 are generally consistent, with presence probabilities exceeding 0.6. Regarding the annual mean temperature (Figure 2b), conditions become increasingly favorable for *E. acuminatum*’s survival when the temperature reaches 13.2 °C. Their peaks all occur between 13.2 °C and 18.5 °C. Bio 3 describes the ratio of diurnal to annual temperature variation (Figure 2c). The output value starts to increase when this ratio reaches 23.7, peaks around 30.0, and then declines. Notably, between 33.0 and 38.0, there is a plateau where the output value stabilizes at approximately 0.8. Seasonal temperature variation is typically described by Bio 4 (Figure 2d). The trend shows that as the coefficient increases, the output value rises rapidly at first, stabilizes briefly, and then declines sharply. After this decline, the output value for PA1 (red line) generally falls below 0.6. Bio 5 (Figure 2e) and Bio 6 (Figure 2f) represent the maximum and minimum temperatures of the hottest and coldest months, respectively. The overall trend of Bio 5 mirrors that of Bio 4, featuring an initial rapid increase, a plateau, and then a decline. Bio 5’s plateau ranges from 26.4 °C to 34.6 °C. Notably, while Bio 6 follows a similar trend, its output value rises more quickly during the initial phase compared to the other two variables. Bio 9 (Figure 2g) and Bio 11 (Figure 2h) both pertain to average temperature, describing the driest quarter and the coldest quarter, respectively. For Bio 9, the output value starts to rise when the average temperature of the driest quarter reaches 4.6 °C. It then declines continuously between 8.4 °C and 9.3 °C, eventually falling below 0.6 and stabilizing around 0.5. Bio 11 exhibits a similar trend, but its output value remains consistently ≥ 0.6. Bio 17 (Figure 2i) is the only variable related to precipitation, representing the amount of precipitation in the driest quarter. Its maximum output value is observed between 50 mm and 90 mm. The remaining two variables are topographic: elevation (Figure 2j) and slope (Figure 2k). They show different trends. As elevation increases, the output value initially demonstrates a positive correlation. As elevation increases to around 700 m, the output value reaches a plateau. It begins to decline once elevation exceeds approximately 1300 m. For slope, the output value decreases with increasing incline, demonstrating a negative correlation. The output value levels off and remains constant when the slope reaches about 15°.

### 2.3. The Potential Suitable Habitat for E. acuminatum (Current)

Figure 3 displays the predicted potential distribution of *E. acuminatum* in contemporary times using two ensemble models: EMca (Figure 3a) and EMwm (Figure 3b). The target species is primarily concentrated in the southwestern region, notably in most of Yunnan and Chongqing, eastern Sichuan, northwestern Hunan and Guangxi, and western Hubei. These regions also correspond to the primary areas of high suitability. In EMca, in addition to the provinces with concentrated distributions, there are also areas of high suitability in Shaanxi, Anhui, Jiangsu, Shanghai, and Guangdong. Overall, the high suitability areas predicted by EMca cover a slightly larger proportion than those predicted by EMwm.

### 2.4. Dynamic Changes in Suitable Habitats for E. acuminatum Under Different Modeling Methods/Year Combinations

This study primarily examines the distribution trends of *E. acuminatum* for the future decades of the 2050s, 2070s, and 2090s, using two integrated modeling methods: EMca and EMwm. It also includes potential distribution maps for *E. acuminatum* (Figure 4), which visually demonstrate the projected changes in the species’ future distribution. Figure 4 shows the changes in suitable area for the three periods using the two integrated modeling methods. Combining Figure 4 and Table 3, there is only a slight difference between the results of the two EM methods, indicating the reliability of the findings. This study examines changes in terms of the areas of loss, stability, and gain. The thresholds for the three regions are 2215 km^2^ to 3418 km^2^, 485,436 km^2^ to 506,489 km^2^, and 12,487 km^2^ to 33,347 km^2^. Among these, the gain area is significantly larger than the loss area. Notably, in the 2090s, both expansion and contraction are most pronounced, with percentage gain ranging from 92.84% to 95.08% and percentage loss ranging from 7.01% to 9.80%.

### 2.5. Migration of Centroid of Suitable Habitats for E. acuminatum

Figure 5 illustrates the movement direction and distance of the distribution centroid for *E. acuminatum* across different periods. Currently, the centroid is located within Guizhou Province. The movement direction is relatively consistent across the three periods, showing an overall shift towards the northeast. In the 2050s, the centroid starts in the northern part of Guizhou Province (107.93° E, 28.73° N), moves through the southeastern region of Chongqing Province, and ultimately reaches the southern part of Hubei Province (109.46° E, 29.60° N), covering a total distance of 176.7 km. In the 2070s and 2090s, the centroid continues to shift northeast, moving an additional 118.2 km (110.37° E, 30.31° N) and 74.4 km (110.98° E, 30.73° N), respectively. By the end of the 2090s, the centroid is located in the southwestern part of Hubei Province.

## 3. Discussion

Many SDMs are designed to predict species distributions, each with its own strengths and limitations. The powerful ensemble modeling platform Biomod2 enables the integration of multiple SDMs (such as MaxEnt, GBM, RF, etc.) into a unified framework [35]. It produces results by integrating different models using methods like weighted averaging. The advantage of this approach is that it combines the strengths of various models, offering a comprehensive and varied perspective for predicting species distributions [36]. The results obtained using this method effectively mitigate the limitations associated with a single model’s bias toward specific datasets or environmental conditions. This study derived the EMca and EMwmean through two integration approaches, both of which passed the TSS, Kappa, and AUC tests with excellent results.

The results of this study indicate that temperature is the primary factor influencing the distribution of *E. acuminatum*. Of the 11 variables examined, 7 are associated with temperature. These variables include annual mean temperature (Bio 1), isothermally (Bio 3), temperature seasonality (Bio 4), maximum temperature of the warmest month (Bio 5), minimum temperature of the coldest month (Bio 6), mean temperature of the driest quarter (Bio 9), and mean temperature of the coldest quarter (Bio 11). Therefore, the interaction between temperature and humidity is a key factor influencing the distribution of *E. acuminatum*. This species is characterized by being shade-loving, cold-tolerant, moisture-loving, and flood-intolerant [37]. Its wild distribution areas are typically located near shaded, moist forest understories and drainage ditches. These areas can maintain soil moisture while promptly draining excess water through the ditches [38]. This aligns with the findings of this study. Among these factors, temperature is the dominant factor influencing the distribution of *E. acuminatum*, which is consistent with broad ecological theory. The impact of temperature on species distribution is one of the fundamental principles in ecology, with many studies indicating that temperature changes directly affect plant growth cycles, reproductive capacity, and survival environments [39]. *E. acuminatum*’s sensitivity to temperature is consistent with its habitat adaptability, especially in shaded, moist forest environments, where the suitable temperature range provides stable growing conditions for the species. Specifically, temperature variables such as annual mean temperature (Bio 1) and minimum temperature of the coldest month (Bio 6) influence the species’ adaptability to the environment and its distribution range. Integrating these findings with broader ecological theory helps us better understand how species respond to environmental changes in the context of climate change and provides crucial information for species conservation and management. For instance, with global warming, the rise in temperature may lead to shifts in the suitable habitat of species, and future conservation measures may need to consider how to maintain habitat suitability under new climate conditions. In terms of species management, temperature sensitivity can serve as an important basis for determining the scope of protected areas and ecological restoration strategies. For example, in areas with higher temperatures, adjustments to cultivation methods or planting locations may be necessary to ensure the species’ growth suitability. Furthermore, combining temperature with other environmental variables, such as water–heat conditions and topographic factors, can provide more comprehensive scientific support for species conservation. This aligns with the study’s findings, suggesting that the model is highly reliable. However, it is crucial to acknowledge that, while water and heat conditions are the primary factors affecting the potential geographic distribution of *E. acuminatum*, topographic constraints should also be considered [40]. Building on this, the study also analyzed the topographic variables Elev, aspect, and slope. Among these, Elev has the highest contribution rate at 10.88% (Table 2). Nonetheless, a species’ distribution is shaped by both biological and abiotic factors. Future predictive efforts should focus on incorporating a wider range of variables to achieve more comprehensive predictions.

For the current suitable distribution of *E. acuminatum*, both integrated models indicate that it is concentrated in southern China. This distribution closely matches the areas covered by the subtropical monsoon climate in China. Most of these areas are also within subtropical evergreen broadleaf forests, which provide the climatic conditions suitable for *E. acuminatum*. Furthermore, the extensive forested areas in these regions align with its preference for shaded, moist environments. This consistency further validates the accuracy of the study. Changes in suitable areas over time are also an intriguing aspect. As shown in Table 3, most areas remain stable. From the 2050s to the 2090s, the total reduction in area is within a normal range of fluctuation. However, the area of habitat expansion for *E. acuminatum* is significantly larger than the area of reduction. The suitable area for *E. acuminatum* is projected to increase from 12,487~12,843 km^2^ in the 2050s to 32,381~33,347 km^2^ in the 2090s, indicating a clear upward trend. Overall, the suitable habitat for *E. acuminatum* is expected to expand in the future. Based on this, in the face of the large-scale destruction of wild *E. acuminatum* resources, it is possible to attempt increasing production in the predicted gain areas through understory biomimetic cultivation techniques. This technology has already been applied to other species within the *Epimedium* genus [41].

## 4. Materials and Methods

### 4.1. Data Occurrence Points and Selection

This study collected species occurrence data for *E. acuminatum* through two approaches (online databases and relevant literature). Global distribution data were obtained from the Global Biodiversity Information Facility (GBIF: https://www.gbif.org, accessed on 21 March 2024). In addition to online databases, data were also sourced from the relevant literature on CNKI (https://www.cnki.net/, accessed on 21 March 2024) and Web of Science (https://www.webofscience.com/, accessed on 21 March 2024), which is a primary source for occurrence points in China. In total, 119 occurrence points of *E. acuminatum* were collected. When occurrence points lacked accurate latitude and longitude information, precise geographical locations were obtained using Google Earth (http://ditu.google.cn, accessed on 21 March 2024). To prevent model overfitting, the “Buffer” function in the “Proximity” tool of ArcGIS v.10.8 was used to eliminate duplicate, incorrect, and ambiguous records within a 5 km radius of selected points for *E. acuminatum*, ensuring that only one distribution point per grid cell was retained. Finally, 112 valid occurrence points were selected and converted into .CSV format, and then organized by species name, longitude, and latitude for future use.

### 4.2. Acquisition and Selection of Environmental Variables

After collecting and filtering the geographic occurrence points, species distribution modeling also requires the acquisition and selection of appropriate environmental variables. This study builds the models using 19 bioclimatic variables (Bio 1-BIO 19) and 3 topographic variables (aspect, elevation, and slope) (Table 4). The current climate data are sourced from the WorldClim version 2.1 global climate database (http://www.worldclim.org/, accessed on 25 March 2024) [42], with values representing averages from 1970 to 2000. Future climate data are used for the periods of the 2050s (averaging 2041–2060), the 2070s (averaging 2061–2080), and the 2090s (averaging 2081–2100). Topographic data are sourced from the National Centers for Environmental Information (NOAA NCEI, https://www.ngdc.noaa.gov/, accessed on 25 March 2024), which include three global elevation datasets (DEM) with a resolution of 2.5 arc-minutes to ensure clear and detailed distribution maps.

The strength of correlations between variables can affect prediction results. Specifically, strong correlations between two variables can lead to model overfitting. Selecting variables is a crucial step in ensuring the accuracy of modeling results. Therefore, before constructing the models, a Pearson correlation analysis was conducted to identify variables with high correlations (|*r*| ≥ 0.8), which were then removed. This approach helps effectively avoid model overfitting. Finally, the remaining environmental variables were used for the subsequent predictive work [43].

### 4.3. Model Building

Combining multiple models for species modeling can help mitigate the limitations of individual models, leading to more reliable predictions. The Biomod2 (4.2-4) software used in this study includes 10 individual models: generalized linear models (GLMs), a gradient boosting model (GBM), classification tree analysis (CTA), a generalized additive model (GAM), multivariate adaptive regression splines (MARS), an artificial neural network (ANN), surface range envelope (SRE), flexible discriminant analysis (FDA), random forest (RF), and maximum entropy (MaxEnt). The 10 individual models mentioned above were utilized to simulate the suitable growth areas for *E. acuminatum*. In the modeling process, 75% of the 112 occurrence points for *E. acuminatum* were randomly chosen as the training dataset, while the remaining 25% were reserved for testing. Additionally, to fulfill Biomod2’s modeling requirements and more accurately simulate actual distribution, 1000 pseudo-absence points were randomly generated. Two different sets of background point data (PA1 and PA2) were employed in the modeling process.

These modeling results must be evaluated before final implementation. Each model’s accuracy is assessed using the True Skill Statistic (TSS), Kappa coefficient, and the Area Under the Curve (AUC) of the ROC curve. The TSS value reflects the net prediction success rate on observed samples, ranging from 0 to 1. A TSS value above 0.7 indicates high prediction accuracy, while a TSS value below 0.5 suggests poor accuracy [44]. The Kappa coefficient quantifies the correlation between model predictions and actual observations, ranging from −1 to 1. A value of 1 represents perfect agreement, while −1 denotes complete disagreement. The AUC ranges from 0 to 1: values between 0 and 0.6 indicate poor predictive performance, 0.6 to 0.8 suggest fair performance, 0.8 to 0.9 reflect good performance, and 0.9 to 1.0 signify excellent performance [45]. Overall, the closer the value is to 1, the better the model fit. This study selects models with TSS > 0.7, Kappa > 0.65, and AUC > 0.9, and then reconstructs the ensemble model (EM) using two integration methods in Biomod2: EMca and EMwmean. These two ensemble models are constructed using the majority voting method (EMca) and the weighted probability method (EMwmean), respectively.

### 4.4. Data Processing and Habitat Suitability Map Visualization

The predictions from the two ensemble models were saved in ASC format and imported into ArcGIS v.10.8 to convert them into raster data. Normalization was conducted using ArcGIS v.10.8, followed by categorizing the distribution values based on the probability assessment methods outlined in the IPCC report. The areas were classified according to the suitability index *P* as follows: high suitable area (0.66~1), medium suitable area (0.33~0.66), low suitable area (0.05~0.33), and non-suitable area (0~0.05). Since a suitability index *p* ≥ 0.05 indicates that the species can survive in these habitats, areas with *p* ≥ 0.05 were classified as suitable habitats. These areas were then overlaid on the administrative map of China for visualization, revealing the current suitable habitats for *E. acuminatum*.

Based on this, SDM_Toolbox_v2.5 was used to identify loss areas (regions currently suitable but projected to become unsuitable in the future), stable areas (regions that remain suitable both now and in the future), and gain areas (regions currently unsuitable but projected to become suitable in the future) for *E. acuminatum* across different time periods.

### 4.5. Centroid Shift

The centroid is the central point of a species’ distribution area and can be visualized as a distinct point on a map. Analyzing centroid movement allows for a more accurate understanding of the target species’ overall geographic distribution trends. Consequently, this study utilizes the mean center function in SDM_Toolbox_v2.5 to obtain and analyze the current and future centroids of *E. acuminatum*.

## 5. Conclusions

This study utilized two ensemble models (EMca and EMwmean) integrated with Biomod2 and combined with 22 environmental variables to successfully simulate the potential geographic distribution of *E. acuminatum* for the present and three future periods (2050s, 2070s, and 2090s). In-depth analysis of the data reveals dynamic changes in suitable habitats for *E. acuminatum* in the future. Currently, high suitability areas are primarily found in the southwestern regions of Yunnan, Chongqing, Sichuan, Hunan, Guangxi, and Hubei. Looking ahead, the overall suitable area is projected to expand, with the expansion rate potentially reaching up to 95.08%. In terms of movement direction, the centroid of *E. acuminatum* is shifting overall toward the northeast. The results indicate that temperature is the most critical factor driving these changes. This information is essential for developing conservation strategies and management measures, as it helps ensure the long-term survival of this medicinal plant and the sustainable supply of medicinal resources. It is hoped that these findings will provide valuable reference data for research in related fields.

## Figures and Tables

**Figure 1 plants-14-01065-f001:**
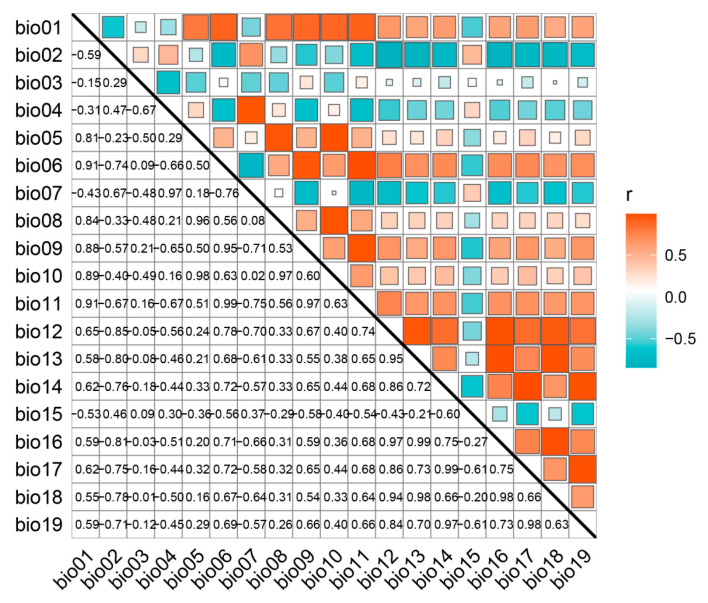
Pearson correlation coefficients of key environmental factors.

**Figure 2 plants-14-01065-f002:**
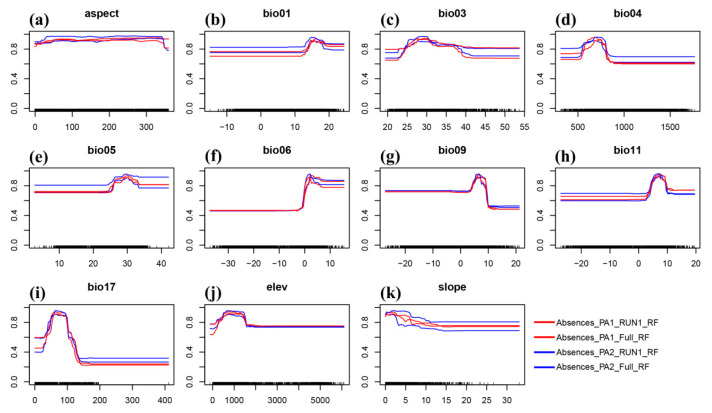
The response curves of the environmental variables that contributed the most to Biomod2. (**a**) Annual mean temperature (Bio 1). (**b**) Isothermally (Bio 3). (**c**) Temperature seasonality (Bio 4). (**d**) Maximum temperature of the warmest month (Bio 5). (**e**) Minimum temperature of the coldest month (Bio 6). (**f**) Mean temperature of the driest quarter (Bio 9). (**g**) Mean temperature of the coldest quarter (Bio 11). (**h**) Precipitation of the driest quarter (Bio 17). (**i**) Elevation (Elev). (**j**) Aspect. (**k**) Slope.

**Figure 3 plants-14-01065-f003:**
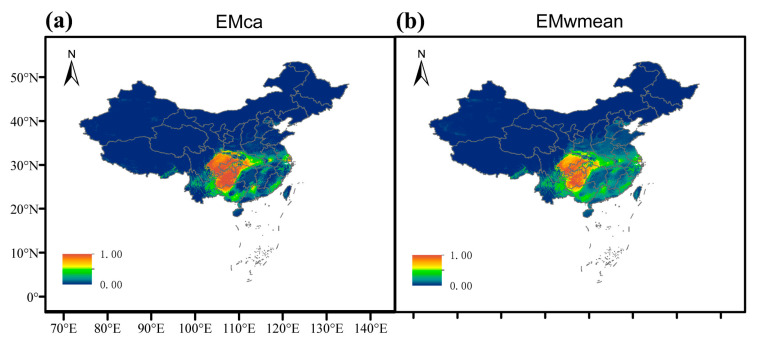
(**a**,**b**) are maps of the potential distribution areas of contemporary *E. acuminatum* in China under two ensemble models (EMca and EMwm), respectively. The probability of *E. acuminatum* is shown by the color scale in the area. Red indicates a highly suitable area with a probability of higher than 0.66, yellow indicates a moderately suitable area with a probability of 0.33–0.66, green indicates a poorly suitable area with a probability ranging from 0.05 to 0.33, and blue represents unsuitable areas.

**Figure 4 plants-14-01065-f004:**
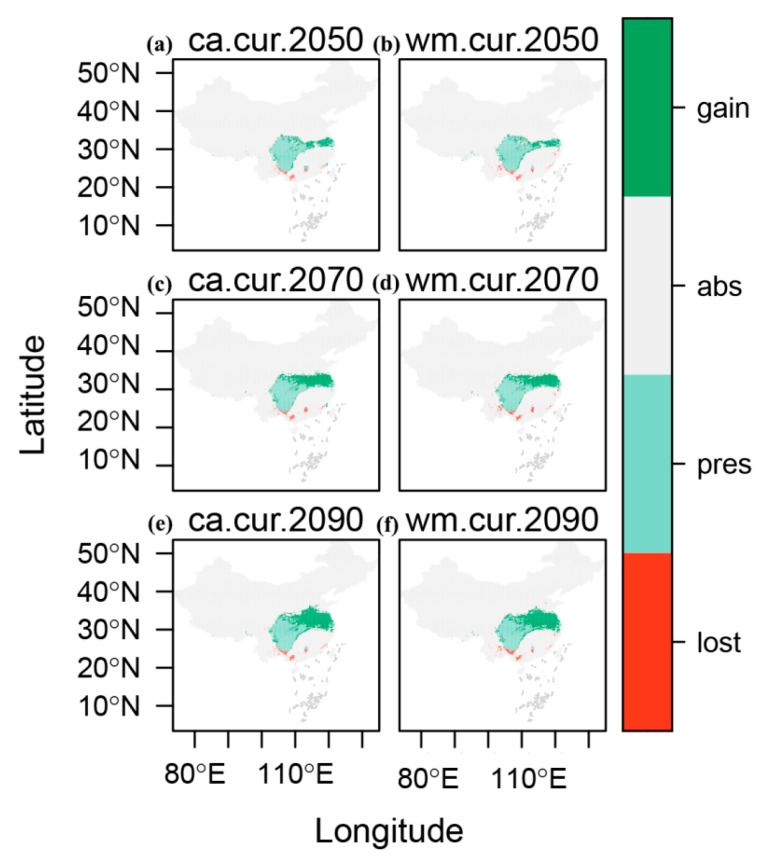
The changes in the suitable habitat area for *E. acuminatum* in China under two ensemble models (EMca and EMwm) and three time periods (2050s, 2070s, and 2090s), including loss, stable, and gain.

**Figure 5 plants-14-01065-f005:**
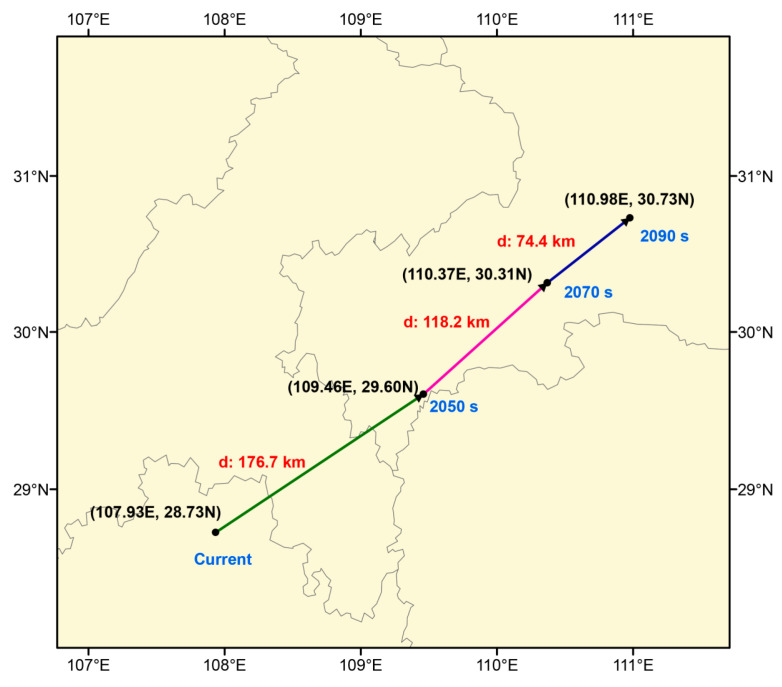
The centroid trajectory and movement distance of *E. acuminatum* in different time periods (2050s, 2070s, and 2090s).

**Table 1 plants-14-01065-t001:** The KAPPA, TSS, and AUC values generated by ten models under different background points.

Model	Data	KAPPA	TSS		AUC
RF	PA1	0.965	0.989		1.000
	PA2	0.965	0.985		0.999
	average value	0.965	0.987		1.000
GAM	PA1	0.917	0.982		0.994
	PA2	0.948	0.989		0.997
	average value	0.948	0.989		0.997
GBM	PA1	0.813	0.948		0.989
	PA2	0.813	0.946		0.988
	average value	0.813	0.947		0.989
ANN	PA1	0.727	0.855		0.967
	PA2	0.810	0.916		0.974
	average value	0.769	0.886		0.971
FDA	PA1	0.761	0.847		0.960
	PA2	0.750	0.848		0.958
	average value	0.756	0.848		0.959
MARS	PA1	0.721	0.866		0.974
	PA2	0.744	0.879		0.971
	average value	0.733	0.873		0.973
GLM	PA1	0.737	0.863		0.975
	PA2	0.735	0.872		0.972
	average value	0.736	0.868		0.974
CTA	PA1	0.686	0.858		0.941
	PA2	0.694	0.868		0.943
	average value	0.690	0.863		0.942
SRE	PA1	0.686	0.705		0.853
	PA2	0.665	0.699		0.850
	average value	0.676	0.702		0.852
MaxEnt	PA1	NA	NA		NA
	PA2	NA	NA		NA
	average value				


**Table 2 plants-14-01065-t002:** Percent contribution (%) of environmental variables in predicting the occurrence of *E. acuminatum* in Biomod2.

Variable	Percent Contribution (%)
Minimum Temperature of the Coldest Month (Bio 6)	64.80
Mean Temperature of the Coldest Quarter (Bio 11)	36.80
Temperature Seasonality (standard deviation × 100) (Bio 4)	33.38
Annual Mean Temperature (Bio 1)	28.69
Precipitation of the Driest Quarter (Bio 17)	27.45
Mean Temperature of the Driest Quarter (Bio 9)	21.55
Maximum Temperature of the Warmest Month (Bio 5)	13.16
Isothermally (Bio 2/BIO 7) (×100) (Bio 3)	11.03
Elevation (Elev)	10.88
Aspect	3.14
Slope (°)	3.05

**Table 3 plants-14-01065-t003:** The area of suitable habitat for *E. acuminatum* in China under two ensemble models (EMca and EMwm) and three time periods (2050s, 2070s, and 2090s), including loss, stable, gain, as well as PercLoss and PercGain.

Period	Model	Loss (km^2^)	Stable (km^2^)	Gain (km^2^)	PercLoss (%)	PercGain (%)
2050s	EMca	2215	505,940	12,843	6.32	36.62
	EMwm	2846	506,489	12,487	8.16	35.80
2070s	EMca	2367	494,820	23,963	6.75	68.32
	EMwm	2862	495,803	23,173	8.21	66.44
2090s	EMca	2458	485,436	33,347	7.01	95.08
	EMwm	3418	486,595	32,381	9.80	92.84

**Table 4 plants-14-01065-t004:** Three terrain factors and 19 bioclimatic variables.

Abbreviation	Description
BIO 1	Annual mean temperature (°C)
BIO 2	Mean diurnal range (mean of monthly (max temp-min temp)) (°C)
BIO 3	Isothermally (Bio 2/BIO 7) × 100
BIO 4	Temperature seasonality (standard deviation × 100) (C of V)
BIO 5	Maximum temperature of the warmest month (°C)
BIO 6	Minimum temperature of the coldest month (°C)
BIO 7	Temperature annual range (Bio 5-BIO 6) (°C)
BIO 8	Mean temperature of the wettest quarter (°C)
BIO 9	Mean temperature of the driest quarter (°C)
BIO 10	Mean temperature of the warmest quarter (°C)
BIO 11	Mean temperature of the coldest quarter (°C)
BIO 12	Annual precipitation (mm)
BIO 13	Precipitation of the wettest month (mm)
BIO 14	Precipitation of the driest month (mm)
BIO 15	Precipitation seasonality (C of V)
BIO 16	Precipitation of the wettest quarter (mm)
BIO 17	Precipitation of the driest quarter (mm)
BIO 18	Precipitation of the warmest quarter (mm)
BIO 19	Precipitation of the coldest quarter (mm)
Elevation (Elev)	Elevation of the terrain
Slope	Slope or obliquity of the terrain
Aspect	The direction or orientation of the earth’s surface

## Data Availability

The data supporting the results are available in a public repository at GBIF.org (21 March 2024) GBIF Occurrence Download https://doi.org/10.15468/dl.48st2b and Zhiling Wang (2024). *Epimedium acuminatum* Franch. figshare. Dataset. https://doi.org/10.6084/m9.figshare.25534711.v1.

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
