# Peer review of "Predicting the Future Geographic Distribution of the Traditional Chinese Medicinal Plant *Epimedium acuminatum* Franch. in China Using Ensemble Models Based on Biomod2"

_plants, 2025, doi:10.3390/plants14071065_

Round 1

Reviewer 1 Report

Comments and Suggestions for Authors

Major comments

  1. In the paragraph on “Epimedium acuminatum,” the authors did not mention any traditional usage of the species or even the antimicrobial uses. The authors need to provide more information about the species, such as known geographic distribution (in China and all over the world, if possible), ecology, height, and elevation.
  2. The results need to follow a sequence. For instance, evaluation of environmental variables should come before SDM evaluation.
  3. What is the basis for using the 19 bioclim and three topographic data?
  4. General editorial work, minor spellings, and space where applicable.
  5. Figure 2 and Table 5 should be moved to the supplementary.
  6. The discussion needs to be compared with previous studies with similar results or distribution.
  7. While field records might not be that important in this kind of study, it is important for the authors to ensure that they consult past literature for more occurrence records. Though 119 seems like a good number to work with.
  8. The authors need to justify using 2.5 arc-min when 0.5 arc-min would give a finer detail and high resolution.
  9. Kindly cite all the packages and software used (such as ArcGIS, Web of Science, and CNKI).
  10. The conclusion needs to be revised to incorporate practical steps toward conserving medicinal plants.
  11. All the species names in the reference section should be in italics.

Minor comments

Line 3: “Epimedium acuminatum (Franch)” should be “Epimedium acuminatum (Franch)” and all other places in the manuscript such as line 10 and 39.

Line 13 – 16: annual mean temperature (Bio 1), there should be space between temperature and (Bio 1), and other bioclim variables mentioned in line 13 – 16. Also, a comma should be used instead of the sign separating the bioclims.

Lines 22 and 23: There is no need to number the keywords.

Line 37: “Epimedium” should be in italics.

Line 56 – 58: Since you mentioned many studies, kindly add more references to buttress your point.

Line 128: “spect” should be “aspect”

Line 155: delete (elev).

Line 189 - 190: Since you’re writing PercGain and PercLoss for the first time, it is important to write it in full to avoid confusion for readers with limited knowledge of this kind of study.

Line 243 – 248: Kindly provide references to back up your claims.

Line 274: Kindly mention the version of WorldClim used and cite the reference.

Line 284 – 290: Kindly provide references to back up your claims. And what software or package did you use for the correlation?

Line 305 – 314: Kindly provide references for the TSS and AUC range to confirm high accuracy.

Line 64: Kindly mention the version of the Biomod2 package and cite as appropriate.

Reviewer 2 Report

Comments and Suggestions for Authors

The manuscript entitled “Predicting the future geographic distribution of traditional Chinese medicinal plant Epimedium acuminatum (Franch) in China using Ensemble Models based on Biomod2” focuses on predicting the potential distribution of this species using the Biomod2 model and 22 bioclimatic variables. The study identifies 11 key environmental variables that affect the plant's habitat suitability, including annual mean temperature, temperature seasonality, and precipitation variables temperature and precipitation It highlights the current high suitability areas primarily across several provinces in China, such as Yunnan and Sichuan. The study emphasizes the importance of sustainable practices for conserving genetic resources of E. acuminatum and highlights the potential threats posed by climate alterations to the species distribution.

The paper would be sufficient to merit publication in Plants, though a revision is recommended, which needs to include the following points:

general suggestion: The English language could be refined a little, grammar and style polished for better readability I suggest either consulting a native speaker or running the text through a program like InstaText or similar.

Introduction:

Lines 26-36  The first paragraph needs to be revised to have a more academic tone. The first sentence is too general, it should be removed

While the introduction touches on the importance of plants in medicine, it could benefit from a more focused discussion on the specific reasons why Epimedium acuminatum is significant. Including more details about its specific medicinal properties or historical uses could enhance its relevance.

The authors should improve the transitions between different sections of the introduction to enhance its clarity.

Lines 53-54  For example, the shift from discussing the threats of relying on wild resources to highlighting the importance of studying geographic distribution could be more clearly emphasized, ensuring a smoother logical flow.

The study's objectives should be clearly outlined at the end of the Introduction to enhance clarity and provide a strong foundation for the research. Additionally, an explanation of how the study contributes to the conservation of E. acuminatum would reinforce its significance.

A brief discussion on the relevance of the chosen model (Biomod2) would further strengthen the rationale behind its use.

Line79 The beginning of the last sentence of the introduction: “The study includes a discussion and summary ... “ should be removed.

Results

The tables are labeled with two numbers (e.g., Table 1-2, Table 2-3 etc.), while the text refers to them as single tables (e.g., Table 1, Table 2, etc.). This inconsistency may cause confusion for readers. Please ensure that table numbering is consistent throughout the manuscript. If two tables are merged into one, clarify this in the text or consider separating them. Additionally, if automatic numbering is used, verify that all references are correctly updated to match the actual table labels.

Line 84 Phrases like "Bulleted lists look like this" seem like an editing oversight and should be removed.

Line 86 While RF is identified as the best-performing model, it would be beneficial to briefly mention why it outperformed the others. Did it have higher sensitivity, better generalization, or was it more robust with the dataset?

Line89 The exclusion of SRE and MaxEnt is mentioned briefly, but there is no explanation of why MaxEnt was not assessed (NA is unclear). Was there an issue with data compatibility, or was it simply not included?

Line 104  "isothermally" should be corrected to "isothermy".

Lines 103-109 The selection of 8 climate and 3 topographic variables is well explained, but additional justification for choosing only these would improve clarity. For example, were less relevant variables removed purely based on Pearson correlation, or was ecological significance also considered?

Lines 108-109 The contribution percentages should be explicitly tied to their ecological importance—what does it mean that Bio 6, Bio 11, and Bio 4 had the highest contributions?

Lines 118–121 The description of response curves is quite basic. Consider briefly explaining how these curves were interpreted—what trends were observed? How do certain variables impact presence probability? Including an example interpretation (e.g., "E. acuminatum shows a higher probability of occurrence in areas with Bio 6 values between X and Y") would make this section more insightful.

Lines 161–169 The discussion of suitable habitat could be improved by adding some comparison between the two ensemble models (EMca and EMwm). Why does EMca predict a slightly larger high-suitability area? If there are significant differences in predictions between the two models, consider discussing which might be more ecologically realistic based on prior studies or expert knowledge

Discussion

The discussion appears to be rather brief, which limits a thorough examination of the study’s findings and their broader implications. Typically, the discussion section should offer a more comprehensive analysis, including comparisons with other studies, a deeper exploration of the limitations, and suggestions for future research. Additionally, only four references are provided, which may not be sufficient to support the arguments presented. I recommend expanding the discussion by including more references and delving deeper into the relevance and implications of the study's results.

My general suggestion to refine the English language throughout the manuscript to improve readability, particularly relates to the discussion section, where clarity and coherence are especially important.

Here are some suggestions that the authors should consider to enhance the quality and depth of the discussion:

  1. The first paragraph of the discussion seems more appropriate for the 'Materials and Methods' section, as it primarily focuses on the technical aspects of the SDM models and the Biomod2 platform used in the study. I recommend moving this paragraph to M&M.
  2. The second paragraph seems more suitable for the 'Results' section, as it primarily presents the study's findings regarding the temperature variables influencing the distribution of acuminatum. In the discussion section, it would be more appropriate to interpret these findings and relate them to broader context.
  3. While the study clearly identifies temperature as a key factor in the species distribution, it would be beneficial for the authors to discuss in more detail how these findings align with broader ecological theories or how they might be applied in species management.
  4. Although the study projects a significant expansion of suitable habitats by the 2090s, a deeper analysis of the potential ecological consequences of this expansion would be valuable. Do the authors foresee any negative effects, such as increased competition with other species or ecosystem changes?

Material and methods

Line 257 The phrase "two approaches" in the first sentence is vague—briefly specify what these two methods are.

Maintain consistent terminology (e.g., sometimes "occurrence points" is used, while other times "distribution records"  and “ distribution points” is mentioned).

Lines 265-267 It is unclear whether duplicate records were removed solely based on spatial proximity or if other factors (e.g., collection date) were considered. Adding more details about the criteria for valid points would be helpful.

Lines 286-288 While Pearson correlation (|r|≥0.8) is mentioned, it would be useful to state how many variables were removed and whether any additional selection was based on ecological relevance rather than just statistical correlation.

Lines 294-298  The authors should use standard English commas instead of Asian-style punctuation marks.

Line 298 The study uses multiple models in Biomod2, but it would be beneficial to briefly explain why these specific models were chosen.

It is not specified whether visual inspection of the data was performed before final modeling.

Lines 336-340 It is unclear how centroid shifts were interpreted—were ecological factors considered, or was it purely a geometric displacement?

Comments on the Quality of English Language

The English language could be refined a little, grammar and style polished for better readability I suggest either consulting a native speaker or running the text through a program like InstaText or similar.

Reviewer 3 Report

Comments and Suggestions for Authors

Predicting the future geographic distribution of the traditional Chinese medicinal plant Epimedium acuminatum Franch. in China using Ensemble Models based on Biomod2

This study uses well-conducted predictive methodology that produces an excellent and data-based prediction of the future distribution of a medicinal plant in China.  The methods are effective in capturing critical factors, and this is technique that can serve as a model for predictive measures to estimate the future distributions of plant species.  The use of English is, overall, good, but there are numerous wording errors that can be corrected before the paper is accepted and published.  Suggested changes in wording are highlighted in red.

The original describing authority (M. A. Franchet) of Epimedium acuminatum is abbreviated as Franch., and I inserted a period after each “Franch” when it appears in the narrative. Epimedium acuminatum Franch.

There are excellent descriptions of Epimedium acuminatum and reference links at the following sites (among many others), and it would be useful to include several in the References to provide easy access to morphological descriptions, ecological, and medical summaries.

Global Biodiversity Information Facility (GBIF). Epimedium acuminatum Franch.

https://www.gbif.org/species/3981102

Kew Royal Botanic Gardens.  Plants of the World Online (POWO). Epimedium acuminatum Franch.

https://powo.science.kew.org/taxon/urn:lsid:ipni.org:names:107259-1

World Flora Online (WFO). Epimedium acuminatum Franch.

https://wfoplantlist.org/taxon/wfo-0000669946-2024-12?page=1

Missouri Botanical Garden. Epimedium acuminatum. https://www.missouribotanicalgarden.org/PlantFinder/PlantFinderDetails.aspx?taxonid=277792&isprofile=1&gen=Epimedium#:~:text=Epimediums%20are%20commonly%20called%20bishop's,believed%20to%20prevent%20female%20conception) .

Throughout the paper numbered references are listed without a space between their bracketed number and the word that precedes it.  I have added a space, which I believe is the correct way the numbers should be presented – for example:  symbolism [1] as opposed to symbolism[1].

Similarly, items that are enclosed by () should also have a space before parenthesis – for example: annual mean temperature (Bio. 1) versus annual mean temperature(Bio. 1).

Scientific names (genus and species) should be italicized, and I have altred the scientific names for genus and species throughout the manuscript – for example:  Epimedium acuminatum versus Epimedium acuminatum.

In the References section, I was unable to open many of the doi links provided – and I added links following the citations for which I could find them.  I indicated those I was unable to verify, and I urge the authors to make certain those references are correctly listed.

In summary, this is a well-researched and documented paper that provides excellent information regarding the future climate-reflecting distribution of this species.  It is an approach that can more widely used, and in that sense is a model.  The paper should be revised and resubmitted prior to publication.

Comments on the narrative are highlighted in red,please see the attachment.

Round 2

Reviewer 2 Report

Comments and Suggestions for Authors

Dear Authors,

I have carefully reviewed your revised manuscript and am pleased to note that you have responded thoroughly to my earlier suggestions. The changes you have made have improved the quality and clarity of the work, and I believe it is ready for publication in Plants.